# The Sedentary Time and Physical Activity Levels on Physical Fitness in the Elderly: A Comparative Cross Sectional Study

**DOI:** 10.3390/ijerph16193697

**Published:** 2019-10-01

**Authors:** Fernanda M. Silva, João Petrica, João Serrano, Rui Paulo, André Ramalho, Dineia Lucas, José Pedro Ferreira, Pedro Duarte-Mendes

**Affiliations:** 1Department of Sports and Well-being, Polytechnic Institute of Castelo Branco, 6000-266 Castelo Branco, Portugal; j.petrica@ipcb.pt (J.P.); j.serrano@ipcb.pt (J.S.); ruipaulo@ipcb.pt (R.P.); andre.ramalho@ipcb.pt (A.R.); lucasdineia@gmail.com (D.L.); pedromendes@ipcb.pt (P.D.-M.); 2University of Coimbra, 3040-248 Coimbra, Portugal; jpl.ferreira.2010@gmail.com; 3Sport, Health & Exercise Research Unit (SHERU), Polytechnic Institute of Castelo Branco, 6000-266 Castelo Branco, Portugal; 4Research Unit for Sport and Physical Activity (CIDAF), University of Coimbra, 3040-248 Coimbra, Portugal

**Keywords:** ageing, sedentary behaviour, health, physical fitness, accelerometry

## Abstract

Background: Ageing is a life-long process characterized by a progressive loss of physical fitness compromising strength, flexibility, and agility. The purpose of this study was to use accelerometry to examine the relationship between sedentary time, light physical activity (LPA), and moderate to vigorous physical activity (MVPA) with the elderly’s physical fitness. Additionally, we aimed to examine the association between the aforementioned variables on older adults who fulfilled global recommendations on physical activity for health and on those who did not fulfil these recommendations. Methods: Eighty-three elderly (mean ± SD: 72.14 ± 5.61 years old) of both genders volunteered to participate in this cross-sectional study, being divided into an active group (n = 53; 71.02 ± 5.27 years old) and an inactive group (n = 30; 74.13 ± 5.72 years old) according to the established guidelines. Sedentary and physical activity times were assessed using an ActiGraph^®^ GT1M accelerometer, whereas physical fitness was evaluated with the Senior Fitness Test. Results: MVPA time was correlated with lower body mass index (BMI) ((r_s_ = −0.218; *p* = 0.048; −0.3 < r ≤ −0.1 (small)) and shorter time to complete the agility test ((r_s_ = −0.367; *p* = 0.001; −0.5 < r ≤ −0.3 (low)). Moreover, MVPA time was positively correlated with aerobic endurance ((r_s_ = 0.397; *p* = 0.000; 0.5 < r ≤ 0.3 (low)) and strength ((r_s_ = 0.243; *p* = 0.027; 0.3 < r ≤ 0.1 (small)). In the inactive group, MVPA time was positively correlated with upper limb flexibility ((rs = 0.400; *p* = 0.028; 0.5 < r ≤ 0.3 (low)); moreover, sedentary time was negatively correlated with upper limb flexibility ((r = −0.443; *p* = 0.014; −0.5 < r ≤ −0.3 (low)), and LPA time was negatively correlated with BMI ((r = −0.423; *p* = 0.020; −0.5 < r ≤ −0.3 (low)). In the active group, MVPA time was correlated with lower BMI ((rs = −0.320; *p* = 0.020; −0.5 < r ≤ −0.3 (low)), and shorter time to complete agility test ((rs = −0.296; *p* = 0.031; −0.3 < r ≤ −0.1 (small)). Conclusions: Our results reinforce the importance of promoting MVPA practice among the elderly, thereby allowing physical fitness maintenance or improvement.

## 1. Introduction

Evidence suggests that the prevalence of many diseases increase with age and are associated with lower levels of physical fitness, namely aerobic endurance, muscular strength, and balance [1]. Lower levels of physical fitness, in turn, are associated with physical disability [2], increased falling risk and fractures [3], and with a decreased quality of life [4]. Therefore, an increase in physical activity levels has been proposed as a relevant strategy to achieve successful ageing [5]. As a matter of fact, regular physical activity is proven to benefit cardiovascular fitness, prevent falls, and improve muscular strength [6], which will translate into the reduction of sarcopenia [7]. In this sense, the World Health Organization (WHO) recommends either a minimum of 150 min of moderate-intensity aerobic physical activity per week or at least 75 min of vigorous-intensity aerobic physical activity per week or an equivalent combination of moderate to vigorous physical activity (MVPA) performed in bouts of at least 10 min each (e.g., 30 min MVPA/day, five times per week or ≥21.4 min MVPA/day, seven times per week) [8]. Individuals who do not achieve the recommended levels of MVPA are considered physically inactive [9]. Evidence shows that the elderly who meet the aforementioned recommendations [8] have higher lower-body muscular endurance and motor coordination [10]. However, it should be noted that, in previous studies, physical activity duration and intensity were based on self-report methods which are known to reduce data accuracy significantly [11]. Conversely, accelerometry is a more accurate method to assess both physical activity duration and intensity [12]. When using accelerometers, authors verified that the physically active elderly have significant higher aerobic endurance and handgrip strength when compared to the physically inactive elderly [13]. Besides, other evidence suggests that the time devoted to MVPA is positively correlated to walking speed and balance [14]. Despite that, light physical activity (LPA) is also associated with physical fitness improvements in the elderly [15]. In other respects, in a cross-sectional study also assessing elderly people, authors verified that those with lower levels of sedentary behaviour and who frequently interrupted sedentary behaviours throughout the day had better physical fitness results when compared to those who practised MVPA [16].

Despite all the evidence supporting the relationship between physical activity and numerous health indicators, including elderly’s physical fitness, sedentary behaviours are considered a new risk factor to health among the elderly, regardless of physical activity levels [17]. According to the most recently updated definition, sedentary behaviours are any waking behaviours characterized by an energy expenditure of ≤1.5 METs, while in a sitting, reclining, or lying posture [18]. Available data suggest a relationship between higher sedentary times and a higher risk of cardiovascular disease and early death [19]. Regarding elderly’s physical fitness, evidence shows that sitting for longer than 4 h/day is a risk factor to men’s losses of strength, flexibility, and aerobic endurance; and women’s losses of balance, strength, agility, walking speed, and aerobic endurance [20]. Since the elderly are typically physically inactive and spend an average of 9.4 h/day in sedentary behaviors [21], it is necessary to emphasize the importance of physical fitness among the elderly. 

Even though some studies focused on the topic, there is still a need to clarify the relationship between physical fitness and sedentary and physical activity in the elderly, using objective measures [17,22,23]. The purpose of this study was to use accelerometry to examine the relationship between sedentary time, LPA, and MVPA with the elderly’s physical fitness. Additionally, we aimed to examine the association between the aforementioned variables on older adults who fulfilled global recommendations on physical activity for health and on those who did not fulfill these recommendations.

## 2. Materials and Methods

### 2.1. Participants

In this quantitative study with a cross-sectional design, participants were recruited through convenience and intentional sampling. A total of 83 participants, aged between 65 to 87 years (72.14 ± 5.61 years old) of both genders (nfemales = 56; nmales = 27), took part in this study. Through accelerometry data, participants were divided into an active group (n = 53), aged between 65 to 84 years (71.02 ± 5.27 years old), which accomplished ≥ 21.4 min of MVPA per day, and into an inactive group (n = 30), aged between 65 to 87 years (74.13 ± 5.72 years old), which did not reach the referred requirements. Participant recruitment was carried out in senior universities and daycare centres. Self-reported sociodemographic and medical history variables were assessed via questionnaires [16]. Age, sex, marital status, educational level, and medical history for hypertension, dyslipidemia, current medication, and any long-standing condition such as diabetes, asthma, cancer, or heart attack and current smoking status were also reported and classified in two categories (no or yes). Inclusion criteria were: (1) age ≥ 65 years or older; (2) non-institutionalized; (3) residents in Castelo Branco district, Portugal; (4) physically independent according to the 12 items of the Composite Physical Functioning Scale whereby participants would be considered physically independent if they achieved a minimum of 14 points [24]; (5) a minimum of three valid days (>10 h of wear time) accelerometer data, including one weekend day; (6) execution of all tests included in the Senior Fitness Test. Regarding exclusion criteria, participants would be excluded if they mentioned serious cardiorespiratory disease and/or cognitive impairments which could preclude them from performing the experimental protocol. All participants were informed about study purpose and signed an informed consent. All procedures were approved by the local Ethics Committee and were conducted in accordance with the declaration of Helsinki for human studies [25].

### 2.2. Instruments

#### 2.2.1. Accelerometry

Physical activity (light, moderate, or vigorous) and sedentary time were assessed using ActiGraph^®^ GT1M Accelerometers (Fort Walton Beach, FL, USA). This is a valid instrument for objectively measuring physical activity frequency, duration, and intensity [26], and the inactive time through the low magnitude values registered [27]. 

#### 2.2.2. Physical Fitness

Physical functional independence of the elderly was assessed with the Senior Fitness Test developed by Rikli and Jones [28] and validated for the Portuguese population by Baptista and Sardinha [29]. This test battery was designed considering two main purposes: (1) to be easily applied; (2) to meet scientific standards of reliability and validity [30].

### 2.3. Procedures

Accelerometers were programmed with the ActiLife Lifestyle software (v. 4.0, Fort Walton Beach, FL, USA) and activated on the first wearing day at 5:00 am. Data were recorded in 15 s epochs since shorter epochs may more accurately estimate physical activity intensity [31]. Participants were instructed to wear the device on their waists near the right iliac crest during waking hours for 5 consecutive days (3 weekdays and 2 weekend days). Accelerometer data were then analysed using an automated data reduction program (MAHUffe; www.mrc-epid.cam.ac.uk), which was useful for data sorting. Aside from the non-wear time related to water activities or sleeping hours, if no counts/min were registered for ≥60 consecutive min, that period was also considered as non-wear time.

A valid day was considered as 600 min (10 h) of wear time [31]. Data of at least 3 valid days (2 weekdays and 1 weekend day) were considered for analysis. Physical activity intensity was estimated using the following cut-off points: sedentary activity (<100 counts/min); light physical activity (100–2019 counts/min); moderate physical activity (2020–5998 counts/min); vigorous physical activity (≥5999 counts/min) [32]. Physical activity parameters including frequency, duration, and intensity were analysed according to Global Recommendations on Physical Activity for Health [8], which allowed the classification of each participant as physically active (accomplished ≥ 21.4 min of MVPA per day) or inactive (did not accomplish ≥ 21.4 min of MVPA per day). 

The Senior Fitness Test assesses five physical fitness components (aerobic endurance, muscular strength, agility/dynamic balance, flexibility, and BMI), and includes six tests: 6-min walk test (metres); chair stand test (repetitions/30 s); arm curl test (repetitions/30 s); 2.44 m up and go test (seconds); chair sit and reach test (centimetres); and back scratch test (centimetres). Assessments were made in random assigned order and participants were instructed to orderly move from a station to another. All tests were administered by experienced raters and conducted in the morning period.

### 2.4. Statistical Analysis

Descriptive statistics (mean ± standard deviation) were performed for all variables under analysis. Data normality was tested using the Kolmogorov–Smirnov test. Spearman’s rank and Pearson’s coefficients (r) were used for bivariate correlations analysis. Additionally, the coefficient of determination was calculated (r^2^). The strength of the relationship was classified as follows [33]: very high (0.90 < r < 1.00); high (0.70 < r < 0.90); moderate (0.50 < r < 0.70); low (0.30 < r < 0.50); little (0.10 < r < 0.30). All statistical analysis was performed using SPSS software v. 25.0 (IBM, Chicago, Illinois, USA), and the significance level was set at *p* ≤ 0.05. A power analysis using the G*Power (3.1.9.2) computer program indicated that the total sample of 84 people would be needed to detect a medium effect (r = 0.3) [34] with 80% power using the test correlation bivariate normal model, with alfa at 0.05.

## 3. Results

Descriptive statistics of demographic and medical history variables, sedentary time, physical activity levels, and physical fitness are presented on Table 1. It can be noticed that sedentary behaviours represented a great part of the participants day. Concerning physical activity level, on average, they spent more time practising light physical activity rather than MVPA. The participants’ values of BMI are above the recommended level, especially in the inactive group. Regarding physical fitness, when compared to the inactive group, the active group achieved higher average results on aerobic fitness (6 min walk test), upper and lower limb strength (arm curl and chair stand test, respectively), agility/dynamic balance (2.44 m up and go test), and upper and lower limb flexibility (back scratch test and chair sit and reach test).

Table 2 displays the correlations between sedentary and physical activity times (both light and MVPA) with the physical fitness tests. There was a significant correlation between MVPA and all variables under analysis, except with the chair stand, chair sit and reach, and back scratch tests. However, sedentary time and light physical activity were not significantly correlated. Furthermore, there was a negative and little correlation between MVPA time and BMI (rs = −0.218; *p* = 0.048; r^2^ = 6.2%), and a negative and low correlation between MVPA time and agility/dynamic balance (2.44 m up and go test time) (rs = −0.367; *p* = 0.001; r^2^ = 8.6%). Moreover, there was a positive and low correlation between MVPA time and aerobic endurance (6 min walk test) (rs = 0.397; *p* = 0.000; r^2^ = 10.6%), and a positive and little correlation between MVPA time and upper limb strength (arm curl test) (rs = 0.243; *p* = 0.027; r^2^ = 4%).

Table 3 displays the correlations between sedentary and physical activity times with the physical fitness tests on older adults who fulfilled recommendations and on those who did not fulfil. In the inactive group, there was a positive and low correlation between MVPA time and upper limb flexibility (back stretch test) (rs = 0.400; *p* = 0.028; r^2^ = 16.1%). Moreover, there was a negative and low correlation between sedentary time and upper limb flexibility (r = −0.443; *p* = 0.014; r^2^ = 19.6%). Lastly, LPA time showed a negative and low correlation with BMI (r = −0.423; *p* = 0.020; r^2^ = 17.9%). In the active group, there was a negative and low correlation between MVPA time and BMI (rs = −0.320; *p* = 0.020; r^2^ = 9.3%), and a negative and little correlation between MVPA time and agility/dynamic balance (2.44 m up and go test time) (rs = −0.296; *p* = 0.031; r^2^ = 4.5%). 

## 4. Discussion

Since the elderly are typically physically inactive and spend more time in sedentary behaviours, it is necessary to emphasise the importance of physical fitness among this age group. Therefore, the purpose of this study was to use accelerometry to examine the relationship between sedentary time, LPA, and MVPA with the elderly’s physical fitness. Additionally, we aimed to examine the association between the aforementioned variables on older adults who fulfilled global recommendations on physical activity for health and on those who did not fulfil these recommendations.

Regarding correlation analysis between sedentary time, LPA, and MVPA with elderly’s physical fitness, the main outcomes suggest that MVPA is associated with a number of physical fitness components in the elderly, particularly with lower BMI and shorter time to complete the agility/dynamic balance test, and with the increased aerobic endurance and muscular strength. Conversely, the results have shown that LPA and sedentary time are not significantly correlated to any physical fitness component in the elderly. Our results support previous research [14,22,35,36,37,38,39], where MVPA was also positively correlated with aerobic endurance, muscular strength, and agility/dynamic balance. Since physical fitness is essential to perform the activities of daily living (ADL’s) and to reduce the risk of falling in the elderly [40], the aforementioned associations should be taken into consideration. Moreover, higher physical fitness levels, particularly aerobic endurance, have been related to a reduced risk for all-cause mortality and cardiovascular events [41]. Some authors noted that physical activity levels were positively related to functional performance, thereby showing that even low-intensity LPA may benefit physical functioning [42,43]. Conversely, our results do not support significant correlations between LPA and physical fitness components in the elderly. In other studies where the referred association was not verified, the authors stated that, although LPA may be enough to improve mental health, it is not sufficient to significantly improve physical fitness components [14]. Consequently, we tend to agree that physical activity practice improves physical fitness, with MVPA contributing more significantly to that improvement [28]. 

Still, according to our pooled results, we also found that with sedentary time there was no significant correlation with any variable under analysis. However, recent researches have shown that sedentary time is negatively correlated with physical fitness in the elderly [17,22,36,44,45,46]. These results have major public health implications since the decline of physical fitness is associated with physical incapacity, morbidity, and other adverse health consequences [47], which entails a substantial increase in medical costs. Differences between studies outcomes may partly be related to varied health status among participants, e.g., participants groups, including subjects with chronic diseases [45], vulnerabilities, or living in nursing homes [44]. Also, differences in methodologies used to assess physical activity and physical fitness (self-report or objective), can justify these differences [46]. Some of the previous studies that have shown negative associations between sedentary time with the components of physical fitness have adjusted the analysis by MVPA time [17,36,45]. Also, in a cross-sectional study, sedentary time was not significantly associated with impaired muscle strength or gait/mobility in Australian adults aged 36–80 years [48]. Although no statistically significant associations were observed in our sample of older adults, given the potential cardiovascular and metabolic benefits of reducing sedentary time [49], the suggestion to reduce prolonged and excessive sitting time is, in fact, relevant [50].

Furthermore, when analysing the association between the aforementioned variables on older adults who fulfilled global recommendations on physical activity for health (active group) and on those who did not fulfil these recommendations (inactive group), we verified that in the inactive group, MVPA time is positively associated with upper limb flexibility, and sedentary time is negatively associated with upper limb flexibility. Also, LPA is negatively associated with BMI. In the active group, MVPA is associated with lower BMI and shorter time to complete the agility/dynamic balance test. These results suggest that even those who do not meet the recommendations for physical activity for health may achieve some improvements in some physical fitness components. In fact, a recent study showed that even only 5 min/day of MVPA is associated with better physical performance [36]. This result supports the general thought that some physical activity is better than none, even if the health status prevents a person from achieving the recommended goals [51,52]. However, the results of the active group are also important and suggest the importance of fulfilling the guidelines. In this sense, evidence shows that the elderly who meet the aforementioned recommendations have higher lower body muscular endurance and motor coordination [10], and significantly higher aerobic endurance and handgrip strength when compared to the physically inactive elderly [13].

In general, authors recommend developing strategies to reduce sedentary time and to increase MVPA practice among this age group [22,53,54]. Supplementary analyses carried out in some studies suggest a frequent interruption of sedentary behaviours [17,54] and to replace a considerable part of the sedentary time with LPA or MVPA in order to positively influence physical fitness components [14].

The data presented in the current study reveals certain practical applications. Sports and health professionals should reinforce the importance of maintaining or improving physical fitness. That reinforcement can be achieved through recommending regular physical activity practice among the elderly (e.g., avoid transport and walk; climb stairs and avoid the lift; engage in a regular exercise program). There are some limitations to this study. First, besides underestimating upper body movements and activities such as carrying heavy loads, weight training, and cycling [55], accelerometer devices are not water-resistant. Furthermore, they cannot capture postural information (i.e., sitting vs. standing still), which may lead to the overestimation of sedentary time [18]. Secondly, the cross-sectional study design does not allow for causal conclusions to be drawn. Also, this was a non-random sample, and the accelerometers were not worn for seven days. This is an important issue because behaviour patterns may reflect physical fitness. Future longitudinal or experimental studies are needed to address this limitation. Also, it would be relevant to analyse not only the amount of physical activity but also the type of physical activity performed (e.g., using the Physical Activity Questionnaire [56]).

## 5. Conclusions

Our findings suggested there was a negative and little correlation between MVPA time and BMI, and a negative and low correlation between MVPA time and agility/dynamic balance (2.44m up and go test time). Moreover, there was a positive and low correlation between MVPA time and aerobic endurance (6 min walk test), and a positive and little correlation between MVPA time and upper limb strength (arm curl test). According to the results between the group analysis (active and inactive), significant correlations were found. In the inactive group, there was a positive and low correlation between MVPA time and upper limb flexibility (back stretch test) and a negative and low correlation between sedentary time and upper limb flexibility. LPA times show a negative and low correlation with BMI. In the active group, there was a negative and low correlation between MVPA time and BMI and a negative and little correlation between MVPA time and agility/dynamic balance. In conclusion, our results reinforce the importance of promoting MVPA practice among the elderly, thereby allowing physical fitness maintenance or improvement. Future prospective studies using objective assessment of physical activity are required to clarify causal relationships between physical activity levels and physical fitness among older people. For the maximized benefits of physical activity, the elderly should be encouraged to interrupt the daily sedentary behaviour, avoiding long sitting periods.

## Figures and Tables

**Table 1 ijerph-16-03697-t001:** Descriptive statistics and normality of distribution of demographic and medical history variables, sedentary time, physical activity levels, and physical fitness across all samples and within groups (active and inactive).

	Total (n = 83)	Inactive Group (n = 30)	Active Group (n = 53)
Demographic characteristics			
Age, years	72.14 ± 5.61	74.13 ± 5.72	71.02 ± 5.27
Women, (%)	67.5	76.7	62.3
College education or higher %	20.48	24.14	22.22
Married (%)	51.81	56.67	60.47
BMI (kg/m^2^)	28.52 ± 4.02	29.15 ± 4.17	28.17 ± 3.93
Medical history			
Diabetes (%)	13.9	17.24	11.63
Asthma (%)	5.4	6.9	4.44
History of Cancer (%)	9.5	10.34	8.89
Heart Attack (%)	5.5	7.14	4.44
Hypertension (%)	42.5	51.72	36.36
Dyslipidemia (%)	9.4	20.0	2.56
Others diseases (%)	31.3	36.0	28.29
Take medication (%)	83.1	93.3	75.60
Current smokers (%)	1.4	0	2.32
Physical activity variables ^a^			
Wear time (min/d)	782.47 ± 80.59	756.09 ± 70.83	797.40 ± 82.55
Sedentary (min/d)	458.10 ± 78.68	462.11 ± 74.99	455.83 ± 81.31
LPA (min/d)	291.16 ± 91.20 *	284.10 ± 98.51	295.16 ± 87.51 *
MVPA (min/d)	33.46 ± 27.25 *	9.88 ± 5.58 *	46.81 ± 25.52 *
Physical performance variables			
6 min walk test (m)	482.25 ± 98.60	437.33 ± 114.62	507.68 ± 78.61
Chair stand test (reps/30s)	15.04 ± 5.06 *	14.23 ± 4.96	15.49 ± 5.10 *
Arm curl test (reps/30s)	20.07 ± 6.69 *	17.67 ± 5.57 *	21.43 ± 6.93
2.44 m up and go test (s)	6.22 ± 2.25 *	6.97 ± 2.55 *	5.79 ± 1.95 *
Chair sit and reach test (cm)	−0.90 ± 7.34 *	−1.80 ± 7.18 *	−0.40 ± 7.44 *
Back scratch test (cm)	−11.28 ± 8.21	−13.16 ± 7.73	−10.21 ± 8.35

* *p* < 0.05—Data are not normally distributed. Note: n, subjects number; min/d, minutes per day; LPA, light physical activity; MVPA, moderate to vigorous physical activity; BMI, body max index; min, minutes; m, metres; reps, repetitions; s, seconds; cm, centimetres; ^a^ Accelerometry—2 week valid days and 1 weekend valid day.

**Table 2 ijerph-16-03697-t002:** Bivariate correlation between participants sedentary time and physical activity levels with physical fitness.

		BMI	6 min Walk Test	Chair Stand Test	Arm Curl Test	2.44 m Up and Go Test	Chair Sit and Reach Test	Back Stretch Test
**MVPA (min/d)**	*r*	−0.218 ^a^*	0.397 ^a^**	0.163 ^a^	0.243 ^a^*	−0.367 ^a^**	0.032 ^a^	0.211 ^a^
	*p*	0.048	0.000	0.142	0.027	0.001	0.772	0.056
Sedentary (min/d)	*r*	0.146 ^b^	−0.182 ^b^	0.167 ^a^	0.124 ^a^	0.065 ^a^	0.112 ^a^	−0.111 ^b^
	*p*	0.187	0.099	0.131	0.264	0.562	0.312	0.318
LPA (min/d)	*r*	−0.157 ^a^	0.032 ^a^	−0.083 ^a^	−0.069 ^a^	0.044 ^a^	−0.030 ^a^	0.148 ^a^
	*p*	0.157	0.772	0.458	0.538	0.693	0.790	0.182

* Correlation is significant at the 0.05 level. ** Correlation is significant at the 0.01 level. ^a,b^ Spearman^a^ or Pearson^b^ correlation according to distribution normality. Note: MVPA, moderate to vigorous physical activity; LPA, light physical activity; min/d, minutes per day; BMI, body max index; min, minutes; m, metres.

**Table 3 ijerph-16-03697-t003:** Bivariate correlation between sedentary time and physical activity levels with physical fitness on older adults who fulfilled global recommendations on physical activity for health (active group) and on those who did not fulfil these recommendations (inactive group).

	Inactive Group (n = 30)		Active Group (n = 53)	
	*r*	*p*	*r*	*p*
MVPA (min/d)				
BMI	0.122 ^a^	0.519	−0.320 ^a^*	0.020
6 min walk test	0.282 ^a^	0.131	0.259 ^a^	0.061
Chair stand test	0.175 ^a^	0.356	0.109 ^a^	0.437
Arm curl test	0.066 ^a^	0.731	0.062 ^a^	0.658
2.44 m up and go test	−0.144 ^a^	0.449	−0.296 ^a^*	0.031
Chair sit and reach test	0.203 ^a^	0.282	−0.173 ^a^	0.215
Back scratch test	0.400 ^a^*	0.028	0.044 ^a^	0.755
Sedentary (min/d)				
BMI	0.295 ^b^	0.113	0.060 ^b^	0.670
6 min walk test	−0.328 ^b^	0.077	−0.077 ^b^	0.584
Chair stand test	0.225 ^b^	0.231	0.254 ^a^	0.066
Arm curl test	−0.140 ^a^	0.461	0.212 ^b^	0.128
2.44 m up and go test	0.160 ^a^	0.397	0.012 ^a^	0.933
Chair sit and reach test	−0.025 ^a^	0.896	0.177 ^a^	0.205
Back scratch test	−0.443 ^b^*	0.014	0.054 ^b^	0.699
LPA (min/d)				
BMI	−0.423 ^b^*	0.020	−0.014 ^a^	0.924
6 min walk test	0.127 ^b^	0.504	−0.046 ^a^	0.744
Chair stand test	−0.253 ^b^	0.178	−0.004 ^a^	0.978
Arm curl test	−0.010 ^a^	0.959	−0.126 ^a^	0.371
2.44 m up and go test	0.059 ^a^	0.759	0.070 ^a^	0.620
Chair sit and reach test	0.217 ^a^	0.249	−0.195 ^a^	0.161
Back scratch test	0.299 ^b^	0.109	0.033 ^a^	0.816

* Correlation is significant at the 0.05 level. ** Correlation is significant at the 0.01 level. ^a,b^ Spearman^a^ or Pearson^b^ correlation according to distribution normality. Note: MVPA, moderate to vigorous physical activity; LPA, light physical activity; min/d, minutes per day; BMI, body max index; min, minutes; m, metres.

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
