# Peer review of "The Sedentary Time and Physical Activity Levels on Physical Fitness in the Elderly: A Comparative Cross Sectional Study"

_ijerph, 2019, doi:10.3390/ijerph16193697_

Round 1
Reviewer 1 Report
Questions
(1) The most important question is sampling:
A total of 83 participants (72.14±5.61 years old) of both 80 genders (females=56; males=27)
* the youngest age ?
* the oldest age ?
* the age distribution of the two groups? (for example, more young elderly participants in active group, and more older participates in inactive group. )
* Females participants much more than male participants (more than 2 times), why do you not to select the participants with the same sample size? (need explanation, your reasonable thinking.)
(2) participants’ health situation check before the study (the base situation of health)
I think before the study, researchers need to provide the basic information about the participants’ health situation. Physical activity of one person is based on his/her health situation particularly for the elderly. If you can not provide the vital information, we can not judge the net-effect of your study.
(3) Table 3. Mann-Whitney U test results for comparison between groups on chair stand, arm curl, 2.44 179 m up and go, and chair sit and reach tests.
Chair sit and reach (cm), Active group: mean ± SD (-0.91±8.15)
Inactive group: mean ± SD (-0.90±6.63)
The SD is much higher than mean, why? I doubt the measure result.
Author Response
International Journal of Environmental Research and Public Health
Reviser 1
RE: The impact of sedentary time and physical activity levels on physical fitness in the elderly: A cross sectional study
My colleagues and I would like to thank you for the opportunity to resubmit our manuscript to the International Journal of Environmental Research and Public Health. We found that the reviewers’ comments were very helpful and we have done our best to incorporate all of their suggestions and reply to the reviewers` comments. We believe that this has made a significant contribution to the overall quality of the manuscript.
The reviewers’ comments and our actions are attached at the bottom of this letter. We have also included an updated version of our manuscript with all the changes highlighted in yellow.
If you require any additional information, please do not hesitate to get in touch with us.
Reviewer 1 Comments
Comments, Suggestions and Questions for Authors
(1) The most important question is sampling:
A total of 83 participants (72.14±5.61 years old) of both 80 genders(females=56; males=27)
* the youngest age ?
* the oldest age ?
* the age distribution of the two groups? (for example, more young elderly participants in active group, and more older participates in inactive group. )
* Females participants much more than male participants (more than 2 times), why do you not to select the participants with the same sample size?
(need explanation, your reasonable thinking.)
R: The youngest age and oldest age were 65 and 87 years old, respectively. Regarding the age distribution of the two groups, the youngest age in two groups was 65 years old, and the oldest age was 84 years old in the active group, and 87 years old in the inactive group. Since it is a cross-sectional study that aims to portray the reality of a certain moment in the district of Castelo Branco and not a laboratory domain study, participants were selected according to availability and convenience. Women were more available to participate in our study.
Action:
Line 86 – “participants (72.14±5.61 years old)” suggest changing to “aged between 65 to 87 years (72.14±5.61 years old)”
Line 88 – “active group (n=53) (71.02±5.27 years old) suggest changing to “active group (n=53) age between 65 to 84 (71.02±5.27 years old)”
Line 90 – “inactive group (n=30) (74.13±5.72 years old)” suggest changing to “inactive group (n=30) age between 65 to 87 years (74.13±5.72 years old)”.
(2) participants’ health situation check before the study (the base situation of health). I think before the study, researchers need to provide the basic information about the participants’ health situation. Physical activity of one person is based on his/her health situation particularly for the elderly. If you can not provide the vital information, we can not judge the net-effect of your study.
R: We include questions about sociodemographic and medical history variables. It’s important to mention that participants would be excluded if they mentioned serious cardiorespiratory disease and/or cognitive impairments which could preclude them to perform the experimental protocol. They also had to be physically independent according to the 12 items of the Composite Physical Functioning Scale.
Action:
Line 92: introduced “Self-reported sociodemographic and medical history variables were assessed via questionnaires [16]. Age, sex, maritat status, educational level, and medical history for hypertension, dyslipidemia, current medication, and any long-standing condition such as diabetes, asthma, cancer, or heart attack and current smoking status were also reported and classified in two categories (no or yes).”
(3) Table 3. Mann-Whitney U test results for comparison between groups on chair stand, arm curl, 2.44 m up and go, and chair sit and reach tests. Chair sit and reach (cm), Active group: mean ± SD (-0.91±8.15), Inactive group: mean ± SD (-0.90±6.63). The SD is much higher than mean, why? I doubt the measure result.
R: The protocol was performed following all the criteria of the Senior Fitness Test (Rikli & Jones, 1999). When we obtained our results we compared them with other studies, and we realized that some of them obtained identical data (Santos et al., 2012; Sardinha et al., 2015; Jantunen et al., 2017).
References:
(Santos, D.; Silva, A.; Baptista, F.; Santos, R.; Vale, S.; Mota, J., Sardinha, L. Sedentary behavior and physical activity are independently related to functional fitness in older adults. Exp Gerontol 2012, 47, 908-912. doi: http://dx.doi.org/10.1016/j.exger.2012.07.011)
(Sardinha, L.B.; Santos, D.A.; Silva, A.M.; Baptista, F.; Owen, N. Breaking-up sedentary time is associated with physical function in older adults. J Gerontol A Biol Sci Med Sci 2015, 70, 119–24.)
(Jantunen, H.; Wasenius, N.; Salonen, M.; Perala, M.; Osmond, C.; Kautianen, H.; Simonen, M.; Pohjolainen, P.; Kajantie, E.; Rantanen, T. et al. Objectively measured physical activity and physical performance in old age. Age Ageing 2017, 46, 232–237.doi: 10.1093/ageing/afw194)

Reviewer 2 Report
I am afraid this paper needs to be thoroughly revised. These are some of the issues that need to be considered:
The wording in the title "impact of" suggests that a causal relationship could be established, which is not possible in a cross-sectional study.
The aim to "assess the effects of the implementation of Global Recommendations on Physical Activity for Health” is not possible with the chosen study design. This might just be a semantic error though. I guess that the authors mean to assess if older adults who fulfil the recommendations differ in physical fitness from those who do not fulfil the recommendations. Even so, the design does not allow this either. The recommendations are to achieve 150 min per week in MVPA and if the aim to assess this the study participants should have worn the accelerometers for one week, which also is the standard procedure. If a whole week is not available, 22 min/day corresponds to 150 min/week, and the sample should accordingly be divided into those who accomplished 22 min or not. I suggest that the new calculations are performed.
Some of the references in the introduction need to be exchanged to better reflect the current literature and to refer to the original studies, for example ref 7.
Tables 3 and 4 present the same data as table 1, and it would be useful if Table 1 also included, age, sex, wear time and number of valid days.
The discussion needs to be more stringent.
Some statements need to be toned down, such as: “there is clear evidence” on line 38, “ the elderly are not involved in physical activity at a sufficient level to ensure positive outcomes on their health” on line 58, “the reduction of sedentary behaviours seems to be mandatory”.
Th conclusion should only include the results that are statistically significant.
Ref 55 is missing.
Author Response
International Journal of Environmental Research and Public Health
Reviser 2
RE: The impact of sedentary time and physical activity levels on physical fitness in the elderly: A cross sectional study
My colleagues and I would like to thank you for the opportunity to resubmit our manuscript to the International Journal of Environmental Research and Public Health. We found that the reviewers’ comments were very helpful and we have done our best to incorporate all of their suggestions and reply to the reviewers` comments. We believe that this has made a significant contribution to the overall quality of the manuscript.
The reviewers’ comments and our actions are attached at the bottom of this letter. We have also included an updated version of our manuscript with all the changes highlighted in yellow.
If you require any additional information, please do not hesitate to get in touch with us.
Reviewer 2 Comments
Comments and Suggestions for Authors
I am afraid this paper needs to be thoroughly revised. These are some of the issues that need to be considered:
The wording in the title "impact of" suggests that a causal relationship could be established, which is not possible in a cross-sectional study.
R: The title has been changed to the following.
Action:
Line 2: “The impact of sedentary time and physical activity levels on physical fitness in the elderly: A cross sectional study” suggest changing to “The sedentary time and physical activity levels on physical fitness in the elderly: A comparative cross-sectional study”.
The aim to "assess the effects of the implementation of Global Recommendations on Physical Activity for Health” is not possible with the chosen study design. This might just be a semantic error though. I guess that the authors mean to assess if older adults who fulfil the recommendations differ in physical fitness from those who do not fulfil the recommendations. Even so, the design does not allow this either.
R: Cross-sectional studies are particularly useful in identifying associations between variables. From the verified associations, or not, new studies may emerge with other research designs, such as randomized studies. In this sense, cross-sectional studies have high limitations regarding the establishment of cause and effect between variables. We, therefore, welcome your comment, which is in line with the methodological principles of cross-sectional design studies, and in particular with the definition of the second objective of this study. Thus, the objective of the study will be described more explicitly considering that it is a study that has the potential to verify associations between variables and not their effects. Thus, the first objective of the study was changed as well as the second.
Action:
Line 18 and 79: “The purpose of this study was to verify the relation between sedentary time, light physical activity (LPA) and moderate to vigorous physical activity (MVPA) with the elderly’s physical fitness. Additionally, we aimed to examine the association between the aforementioned variables on older adults who fulfilled global recommendations on physical activity for health and on those who did not fulfill these recommendations.”
The recommendations are to achieve 150 min per week in MVPA and if the aim to assess this the study participants should have worn the accelerometers for one week, which also is the standard procedure. If a whole week is not available, 22 min/day corresponds to 150 min/week, and the sample should accordingly be divided into those who accomplished 22 min or not. I suggest that the new calculations are performed.
R: We have made a new analyze of data in according to the suggestion’s of the revisor 2 considering the active group with 22 min/day and the inactive group less than 22m/day.
Action:
Line 87: “Through accelerometry data, participants were divided into an active group (n=53), age between 65 to 84 years (71.02±5.27 years old), which accomplished ≥21.4 min of MVPA per day, and into an inactive group (n=30) age between 65 to 87 years (74.13±5.72 years old), which did not reach the referred requirements.”
Some of the references in the introduction need to be exchanged to better reflect the current literature and to refer to the original studies, for example ref 7.
R: We have performed a new literature search by adding more recent references.
Action:
Linte 42 to 82: “Evidence suggests that the prevalence of many diseases increase with age and is associated with lower levels of physical fitness, namely aerobic endurance, muscular strength and balance [1]. Lower levels of physical fitness, on its turn, are associated with physical disability [2], increased falling risk and fractures [3] and with a decreased quality of life [4]. Therefore, the rise of physical activity levels has been proposed as a relevant strategy to achieve a successful aging [5]. As a matter of fact, regular physical activity is proven to benefit cardiovascular fitness, prevent falls and improve muscular strength [6], which will translate into the reduction of sarcopenia [7]. In this sense, the World Health Organization [WHO] recommends either a minimum of 150 minutes of moderate intensity aerobic physical activity per week or at least 75 minutes of vigorous intensity aerobic physical activity per week or an equivalent combination of MVPA performed in bouts of at least 10 minutes each (e.g., 30 min MVPA/day, 5 times per week or ≥21.4 min MVPA/day, 7 times per week) [8]. Individuals who do not achieve the recommended levels of MVPA are considered physically inactive [9]. Evidence shows that the elderly who meet the aforementioned recommendations [8], have higher lower body muscular endurance and motor coordination [10]. However, it should be noted that, in previous studies, physical activity duration and intensity were based on self-report methods which are known to significantly reduce data accuracy [11]. Conversely, the accelerometer is a precise method to assess both physical activity duration and intensity [12]. When using accelerometers, authors verified that the physically active elderly have significant higher aerobic endurance and handgrip strength when compared to the physically inactive elderly [13]. Besides, other evidence suggests that the time devoted to MVPA is positively correlated to walking speed and balance [14]. Despite that, LPA is also associated with physical fitness improvements in the elderly [15]. In other respects, in a cross-sectional study also assessing elderly people, authors verified that those with lower levels of sedentary behavior and who frequently interrupted sedentary behaviors throughout the day had better physical fitness adjustments when compared to those who practiced MVPA [16].
Despite all the evidence that supporting the relation between physical activity and numerous health indicators including elderly’s physical fitness, sedentary behaviors are considered a new risk factor to health among the elderly, regardless of physical activity levels [17]. According to the last updated definition, sedentary behaviors are any waking behaviors characterized by an energy expenditure ≤1.5 METs, while in a sitting, reclining, or lying posture [18]. Available data suggests a relation between higher sedentary times and higher risk of cardiovascular disease and early death [19]. Regarding elderly’s physical fitness, evidence shows that being sat for longer than 4 hours/day is a risk factor to men’s losses of strength, flexibility and aerobic endurance; and women’s losses of balance, strength, agility, walking speed and aerobic endurance [20]. Since the elderly are typically physically inactive and spend an average of 9.4 hours/day in sedentary behaviors [21], it is necessary to emphasise the importance of physical fitness among the elderly.
Even though some studies focused on the topic, there is still a need to clarify the relation between physical fitness and sedentary and MVPA times in the elderly, using objective measures [17, 22-23]. The purpose of this study was to verify the relation between sedentary time, LPA and MVPA with the elderly’s physical fitness. Additionally, we aimed to examine the association between the aforementioned variables on older adults who fulfilled global recommendations on physical activity for health and on those who did not fulfill these recommendations.”
Tables 3 and 4 present the same data as table 1, and it would be useful if Table 1 also included, age, sex, wear time and number of valid days.
R: We include demographic and medical history variables in table 1, namely, age, sex, marital status, educational level, and medical history for hypertension, dyslipidemia, current medication, and any long-standing condition such as diabetes, asthma, cancer, or heart attack and current smoking status were also reported and classified in two categories (no or yes). We also introduce wear time of accelerometer.
Action:
Line 161 – table 1.
The discussion needs to be more stringent.
R: Thank you for your suggestion. We improve the discussion.
Action:
Line 204 to 277: “Since the elderly are typically physically inactive and spend more time in sedentary behaviours, it is necessary to emphasise the importance of physical fitness among this age group. Therefore, the purpose of this study was to verify the relation between sedentary time, light physical activity (LPA) and moderate to vigorous physical activity (MVPA) with the elderly’s physical fitness. Additionally, we aimed to examine the association between the aforementioned variables on older adults who fulfilled global recommendations on physical activity for health and on those who did not fulfill these recommendations.
Regarding correlation analysis between sedentary time, LPA and MVPA with elderly’s physical fitness, the main outcomes suggest that MVPA is associated with a number of physical fitness components in the elderly, particularly with lower BMI and shorter time to complete the agility/dynamic balance test, and with the increased aerobic endurance and muscular strength. Conversely, the results have shown that LPA and sedentary time are not significantly correlated to any physical fitness component in the elderly. Our results support previous research [14, 22, 35- 39], where MVPA was also positively correlated with aerobic endurance, muscular strength and agility/dynamic balance. Since physical fitness is essential to perform ADL’s and to reduce the risk of falling in the elderly [40], the aforementioned associations should be taken into consideration. Moreover, higher physical fitness levels, particularly aerobic endurance, have been related to a reduced risk for all-cause mortality and cardiovascular events [41].
Some authors noted that physical activity levels were positively related to functional performance, thereby showing that even low intensity LPA may benefit physical functioning [42, 43]. Conversely, our results do not support significant correlations between LPA and physical fitness components in the elderly. In other studies where the referred association was not verified, authors stated that, although LPA may be enough to improve mental health, it is not sufficient to significantly improve physical fitness components [14]. Consequently, we tend to agree that physical activity practice improves physical fitness, with MVPA contributing more significantly to that improvement [28].
Concerning sedentary time, there also was no significant correlation with any variable under analysis. However, recent researches have shown that sedentary time is negatively correlated with physical fitness in the elderly [17, 22, 36, 44-46]. These results have major public health implications since the decline of physical fitness is associated with physical incapacity, morbidity and other health adverse consequences [47], which entails a substantial increase in medical costs. Differences between studies outcomes may partly be related to varied health status among participants, e.g., participants groups including subjects with chronic diseases [45], vulnerable, or living in nursing homes [44]. Also, differences in methodologies used to assess physical activity and physical fitness (self-report or objective), can justify these differences [46]. Some of the previous studies that have shown negative associations between sedentary time with the components of physical fitness have adjusted the analysis by MVPA time [17, 36, 45]. Also, in a cross-sectional study, sedentary time was not significantly associated with impaired muscle strength or gait/mobility in Australian adults aged 36-80 years [48]. Although no statistically significant associations were observed in our sample of older adults, given the potential cardiovascular and metabolic benefits of reducing sedentary time [49], the suggestion to reduce prolonged and excessive sitting time is in fact relevant [50].
Furthermore, when analysing the association between the aforementioned variables on older adults who fulfilled global recommendations on physical activity for health (active group) and on those who did not fulfill these recommendations (inactive group), we verified that in the inactive group, MVPA time is positively associated with upper limb flexibility, and sedentary time is negatively associated with upper limb flexibility. Also, LPA is negatively associated with BMI. In the active group, MVPA is associated with lower BMI and shorter time to complete agility/dynamic balance test. These results suggest that even those who do not meet the recommendations for physical activity for health may achieve some improvements in some physical fitness components. In fact, a recent study showed that even only 5 min/day of MVPA are associated with better physical performance [36]. This result supports the general thought that some physical activity is better than none, even if the health status prevents a person from achieving the recommended goals [51, 52]. However, the results of the active group are also important and suggest the importance of fulfilling the guidelines. In this sense, evidence shows that the elderly who meet the aforementioned recommendations, have higher lower body muscular endurance and motor coordination [10], and significant higher aerobic endurance and handgrip strength when compared to the physically inactive elderly [13].
In general, authors recommend developing strategies to reduce sedentary time and to increase MVPA practice among this age group [22, 53, 54]. Besides, supplementary analyses carried out in some studies suggest to frequently interrupt sedentary behaviours [17, 54], and to replace a considerable part of the sedentary time with LPA or MVPA in order to positively influence physical fitness components [14].
The data presented in the current study reveals certain practical applications. Sports and health professionals should reinforce the importance of preserving or improving physical fitness. That reinforcement can be achieved through recommending regular MVPA practice among the elderly. There are some limitations in this study. First, besides underestimating upper body movements and activities such as carrying heavy loads, weight training, and cycling [55], accelerometers devices are not water resistant. Furthermore, they cannot capture postural information (i.e., sitting vs. standing still), which may lead to the overestimation of sedentary time [18]. Secondly, the cross-sectional study design does not allow for causal conclusions to be drawn. This in an important issue because behaviour patterns may reflect physical fitness. Future longitudinal or experimental studies are needed to adress this limitation. Also, it would be relevant to analyse not only physical activity quantity but also the type of physical activity performed (e.g., using the Physical Activity Questionnaire [56]).”
Some statements need to be toned down, such as: “there is clear evidence”on line 38.
R: We change the introduction to respond to reviewers' requests.
“the elderly are not involved in physical activity at a sufficient level to ensure positive outcomes on their health” on line 58,
R: We change the introduction to respond to reviewers' requests.
“the reduction of sedentary behaviours seems to be mandatory”.
R: Thank you for the suggestion.
Action:
Line 242: “Although no statistically significant associations were observed in our sample of older adults, given the potential cardiovascular and metabolic benefits of reducing sedentary time [49], the suggestion to reduce prolonged and excessive sitting time is in fact relevant [50].”
The conclusion should only include the results that are statistically significant.
R: Thank you for the suggestion. We improved the conclusions.
Action:
Line 279 to 292: “Our findings suggested there was a negative and little correlation between MVPA time and BMI, and a negative and low correlation between MVPA time and aerobic endurance (6-minute walk test time). Moreover, there was a positive and low correlation between MVPA time and agility/dynamic balance (2.44m up and go test time), and a positive and little correlation between MVPA time and upper limb strength (arm curl test). According to the results between group analysis (active and inactive), significant correlations were found. In the active group there was a positive and low correlation between MVPA time and upper limb flexibility (Back stratch test) and a negative and low correlation between sedentary time and upper limb flexibility. LPA time show a negative and low correlation with BMI. In the active group, there was a negative and low correlation between MVPA time and BMI and a negative and little correlation between MVPA time and agility/dynamic balance. In conclusion, our results reinforce the importance of promoting MVPA practice among the elderly, thereby allowing physical fitness maintenance or improvement. Future prospective studies are, however, required to clarify causal relationships between physical activity levels and physical fitness among older people”.
Ref 55 is missing.
R: The reference was not enumerated.
Action:
Line 439: “56. Baecke, J.A.; Burema, J.; Frijters, J.E. A short questionnaire for the measurement of habitual physical activity in epidemiological studies. Am J Clin Nutr 1982, 36, 936-942. doi: 10.1093/ajcn”

Reviewer 3 Report
The impact of sedentary time and physical activity levels on physical fitness in the elderly: A cross-sectional study
Study summary
This study examined cross-sectional association of sedentary time and physical activity with physical fitness, measured using the Senior’s Fitness Test, in an elderly population. The study found that MVPA was associated with BMI and agility, aerobic endurance and strength. No associations were observed for sedentary time.
While I see the need for more studies investigating this research question, the authors need to be clearer on why this research question is important, as well as how it addresses gaps within the literature. I see that the authors are using objective measures of both exposures and outcomes, which is useful, but many studies have been doing this (particularly in the last few years). The age of the sample is unique as many studies don’t focus exclusively on older adults, so perhaps this needs to be discussed further.
In addition to this, I am concerned that the conclusions drawn from the findings are not justified. It is noted that no statistically significant associations with sedentary time were found, but then it is stated that that the authors could verify that longer sedentary time was associated with lower aerobic endurance, etc. I think the authors can suggest a trend in this direction, and discuss potential reasons why statistical significant was not reached (e.g. power?). For these reasons, I recommend a major revision of the manuscript before considering it for further review. My specific comments are below.
Major Revision requiring attention (see below):
Abstract
Line 22: If mentioning the participants being divided into an active and inactive group, please explain why (i.e., it is one of your research questions).
Line 24: Please specify what established guidelines you are referring to.
Introduction
Line 37: An updated definition and reference are available to explain sedentary behaviour.
Line 69: You mention that it is still necessary to clarify the association in the elderly, using objective measures. However, you need to provide further rationale for why this is necessary. For example, older adults do not self-report their activity time very accurately. There are many references that can support such an argument.
Line 71: The sentence starting on this line needs to be reworded as it does not read very clearly.
Methods
Line 84: “third age universities and elderly day-care centres” is unusual terminology. I would suggest you find the international equivalents for these institutions.
Line 94: Please also include details of your ethics approval
Line 95: Were any other variables collected and used in this study, such as demographics and health variables to help describe your sample?
Results
Line 141: In the results section it would be helpful to know more about your sample in terms of their demographics and health measures. These are particularly useful when examining an outcome such as physical fitness in the elderly, as there could be many reasons for their fitness scores other than their activity and sedentary levels.
Line 172: When discussing your results in text, the usual practice is to include the actual figures in text as well as in the table.
Overall comment: It would be interesting to see the association of PA and sedentary time with the SFT stratified by active/inactive group, although I note that your sample size probably precludes you doing this.
Discussion
Line 186: As with the introduction, make sure this opening sentence really explains the uniqueness and novelty of this study.
Line 190: Here its mentioned that MVPA was associated with a reduction in BMI and execution time for some performance test. The word ‘reduction’ implies a longitudinal association, which cannot be inferred from this study design. I would reword this to make it clear it is a cross-sectional association, e.g., MVPA was associated with a lower BMI and shorter time to complete x tests.
Line 198: Your summary paragraph should mention all of the findings, including the ones surrounding sedentary behaviour.
Line 209: This paragraph around sedentary behaviour needs to be re-drafted. I don’t think you have enough evidence to say that you could verify that longer SB is associated with lower aerobic endurance, etc. Many other studies have observed statistically significant findings, so I do understand and agree with the overall implications of these results, however you need to make clear that your study did not observe statistically significant associations. I would then explain potential reasons why these were not observed. It would also be interesting to know if the effect sizes you observed were similar to other studies, where just the confidence intervals were larger?
Line 221: You mention that you “verified that only a minority of the elderly achieved the widely recommended daily 30 minutes of MVPA.” However, in your sample almost half of the participants reached this recommended level (39 out of 83; 47%).
Line 231: You write “this confirms that physical activity level is associated with physical fitness maintenance or increase.” This sentence suggests a longitudinal association, which was not investigated here. All you can do is speculate and relate to previous evidence, therefore I would change the word “confirms” to “suggests”.
Line 245: In discussing the limitations of the study, please also make suggestions for future research regarding how these limitations could be addressed.
Line 246: With your second limitation, go a step further and explain what implications this limitation of the accelerometers is (i.e., they can underestimate total PA / miss-classify PA, etc).
Overall comment: What about power? Were you powered to detect significant associations? Also, I don’t believe the SFT specifically includes a measure of BMI. It’s an important variables but it should be made clear that this was an addition to the SFT.
Author Response
International Journal of Environmental Research and Public Health
Reviser 3
RE: The impact of sedentary time and physical activity levels on physical fitness in the elderly: A cross sectional study
My colleagues and I would like to thank you for the opportunity to resubmit our manuscript to the International Journal of Environmental Research and Public Health. We found that the reviewers’ comments were very helpful and we have done our best to incorporate all of their suggestions and reply to the reviewers` comments. We believe that this has made a significant contribution to the overall quality of the manuscript.
The reviewers’ comments and our actions are attached at the bottom of this letter. We have also included an updated version of our manuscript with all the changes highlighted in yellow.
If you require any additional information, please do not hesitate to get in touch with us.
Reviewer 3 Comments
Comments and Suggestions for Authors
This study examined cross-sectional association of sedentary time and physical activity with physical fitness, measured using the Senior’s Fitness Test, in an elderly population. The study found that MVPA was associated with BMI and agility, aerobic endurance and strength. No associations were observed for sedentary time.
While I see the need for more studies investigating this research question, the authors need to be clearer on why this research question is important, as well as how it addresses gaps within the literature. I see that the authors are using objective measures of both exposures and outcomes, which is useful, but many studies have been doing this (particularly in the last few years).
The age of the sample is unique as many studies don’t focus exclusively on older adults, so perhaps this needs to be discussed further.
In addition to this, I am concerned that the conclusions drawn from the findings are not justified. It is noted that no statistically significant associations with sedentary time were found, but then it is stated that that the authors could verify that longer sedentary time was associated with lower aerobic endurance, etc. I think the authors can suggest a trend in this direction, and discuss potential reasons why statistical significant was not
reached (e.g. power?). For these reasons, I recommend a major revision of the manuscript before considering it for further review. My specific comments
are below.
R: Thank you for your comments. We decided to improve our introduction, so we also justified the relevance of our study. We have also improved our conclusions.
Action:
Line 55: “However, it should be noted that, in previous studies, physical activity duration and intensity were based on self-report methods which are known to significantly reduce data accuracy [11]”.
Line 74 to 83: “ Since the elderly are typically physically inactive and spend an average of 9.4 hours/day in sedentary behaviors [21], it is necessary to emphasise the importance of physical fitness among the elderly. Even though some studies focused on the topic, there is still a need to clarify the relation between physical fitness and sedentary and MVPA times in the elderly, using objective measures [17, 22-23].”
Major Revision requiring attention (see below):
Abstract
Line 22: If mentioning the participants being divided into an active and inactive group, please explain why (i.e., it is one of your research questions).
R: Initially, we had distributed the participants into two groups based on adherence to at least 30 minutes per day of MVPA. However, according to reviewer 2's comments, we felt that the most appropriate way would be to form the two groups by recording at least 21.4 minutes of MVPA or more per day. World Health Organization [WHO] recommends either a minimum of 150 minutes of moderate-intensity aerobic physical activity per week or at least 75 minutes of vigorous-intensity aerobic physical activity per week or an equivalent combination of MVPA performed in bouts of at least 10 minutes each (e.g., ≥21.4 min MVPA/day, 7 times per week).
Line 24: Please specify what established guidelines you are referring to.
R: The guidelines are the Global Recommendations on Physical Activity for Health, established by the World Health Organization [WHO, 2010].
Introduction
Line 37: An updated definition and reference are available to explain sedentary behaviour.
R: We include the last updated definition that explains sedentary behaviour.
Action:
Line 68: “According to the last updated definition, sedentary behaviors are any waking behaviors characterized by an energy expenditure ≤1.5 METs, while in a sitting, reclining, or lying posture [18].”
Line 69: You mention that it is still necessary to clarify the association in the elderly, using objective measures. However, you need to provide further rationale for why this is necessary. For example, older adults do not selfreport their activity time very accurately. There are many references that can support such an argument.
R: Thank you. We include this information in the introduction.
Action:
Line 55 to 58: “However, it should be noted that, in previous studies, physical activity duration and intensity were based on self-report methods which are known to significantly reduce data accuracy [11]. Conversely, the accelerometer is a precise method to assess both physical activity duration and intensity [12].”
Line 71: The sentence starting on this line needs to be reworded as it does not read very clearly.
R: We changed the sentence.
Action:
Line 79: “The purpose of this study was to verify the relation between sedentary time, LPA and MVPA with the elderly’s physical fitness. Additionally, we aimed to examine the association between the aforementioned variables on older adults who fulfilled global recommendations on physical activity for health and on those who did not fulfill these recommendations.”
Methods
Line 84: “third age universities and elderly day-care centres” is unusual terminology. I would suggest you find the international equivalents for these institutions.
R: We found international equivalents for theses institutions.
Action:
Line 91 - ‘in third age universities and elderly day-care centers’ suggest changing to ‘senior universities and day care center`
Line 94: Please also include details of your ethics approval.
R: All procedures were approved by the local ethics Comitte.
Action:
Line 104: “All procedures were approved by the local ethics Comitte and were conducted in accordance with declaration of Helsinki for human studies [25].”
Line 95: Were any other variables collected and used in this study, such as demographics and health variables to help describe your sample?
R: We include demographic and medical history variables in table 1, namely, age, sex, marital status, educational level, and medical history for hypertension, dyslipidemia, current medication, and any long-standing condition such as diabetes, asthma, cancer, or heart attack and current smoking status were also reported and classified in two categories (no or yes). We also introduce wear time of accelerometer.
Action:
Line 161 – Table 1.
Results
Line 141: In the results section it would be helpful to know more about your sample in terms of their demographics and health measures. These are particularly useful when examining an outcome such as physical fitness in the elderly, as there could be many reasons for their fitness scores other than their activity and sedentary levels.
R: Thank you for your suggestion. We include these results in table 1.
Action:
Line 161 – Table 1.
Line 172: When discussing your results in text, the usual practice is to include the actual figures in text as well as in the table.
R: We include the results in the text.
Action:
Line 186: “Table 3 displays the correlations between sedentary and physical activity times with the physical fitness tests on older adults who fulfilled global recommendations on physical activity for health and on those who did not fulfill these recommendations. In the inactive group, there was a positive and low correlation between MVPA time and upper limb flexibility (Back stratch test) (rs=0.400; p=0.028; r2=16.1%). Moreover, there was a negative and low correlation between sedentary time and upper limb flexibility (r=-0.443; p=0.014; r2=19.6%). Lastly, LPA time show a negative and low correlation with BMI (r=-0.423; p=0.020; r2=17.9%). In the active group, there was a negative and low correlation between MVPA time and BMI (rs=-0.320; p=0.020; r2=9.3%), and a negative and little correlation between MVPA time and agility/dynamic balance (2.44m up and go test time) (rs=-0.296; p=0.031; r2=4.5%).”
Overall comment: It would be interesting to see the association of PA and sedentary time with the SFT stratified by active/inactive group, although I note that your sample size probably precludes you doing this.
R: Based on the comments of reviewer 2, the analysis was performed accordingly. So we checked the association between the aforementioned variables on older adults who fulfilled global recommendations on physical activity for health and on those who did not fulfill these recommendations.
Discussion
Line 186: As with the introduction, make sure this opening sentence really explains the uniqueness and novelty of this study.
R: We include a sentence that explains the uniqueness and novelty of this study.
Action:
Line 204: “Since the elderly are typically physically inactive and spend more time in sedentary behaviours, it is necessary to emphasise the importance of physical fitness among this age group.”
Line 190: Here its mentioned that MVPA was associated with a reduction in BMI and execution time for some performance test. The word ‘reduction’ implies a longitudinal association, which cannot be inferred from this study design. I would reword this to make it clear it is a cross-sectional association, e.g., MVPA was associated with a lower BMI and shorter time to complete x tests.
R: Thanks for the suggestion.
Action:
Line 211: “Regarding correlation analysis between sedentary time, LPA and MVPA with elderly’s physical fitness, the main outcomes suggest that MVPA is associated with a number of physical fitness components in the elderly, particularly with lower BMI and shorter time to complete the agility/dynamic balance test, and with the increased aerobic endurance and muscular strength.”
Line 198: Your summary paragraph should mention all of the findings, including the ones surrounding sedentary behaviour.
R: We mention all of the findings in summary.
Action:
Line 211 to 216: “Regarding correlation analysis between sedentary time, LPA and MVPA with elderly’s physical fitness, the main outcomes suggest that MVPA is associated with a number of physical fitness components in the elderly, particularly with lower BMI and shorter time to complete the agility/dynamic balance test, and with the increased aerobic endurance and muscular strength. Conversely, the results have shown that LPA and sedentary time are not significantly correlated to any physical fitness component in the elderly.”
Line 209: This paragraph around sedentary behaviour needs to be redrafted. I don’t think you have enough evidence to say that you could verify that longer SB is associated with lower aerobic endurance, etc. Many other studies have observed statistically significant findings, so I do understand and agree with the overall implications of these results, however you need to make clear that your study did not observe statistically significant associations. I would then explain potential reasons why these were not observed. It would also be interesting to know if the effect sizes you observed were similar to other studies, where just the confidence intervals were larger?
Action:
Line 234 to 244: “Differences between studies outcomes may partly be related to varied health status among participants, e.g., participants groups including subjects with chronic diseases [45], vulnerable, or living in nursing homes [44]. Also, differences in methodologies used to assess physical activity and physical fitness (self-report or objective), can justify these differences [46]. Some of the previous studies that have shown negative associations between sedentary time with the components of physical fitness have adjusted the analysis by MVPA time [17, 36, 45]. Also, in a cross-sectional study, sedentary time was not significantly associated with impaired muscle strength or gait/mobility in Australian adults aged 36-80 years [48]. Although no statistically significant associations were observed in our sample of older adults, given the potential cardiovascular and metabolic benefits of reducing sedentary time [49], the suggestion to reduce prolonged and excessive sitting time is in fact relevant [50].”
Line 221: You mention that you “verified that only a minority of the elderly achieved the widely recommended daily 30 minutes of MVPA.” However, in your sample almost half of the participants reached this recommended level (39 out of 83; 47%).
R: We decided to remove this sentence because we have performed new data analysis by suggestion of other revisor.
Line 231: You write “this confirms that physical activity level is associated with physical fitness maintenance or increase.” This sentence suggests a longitudinal association, which was not investigated here. All you can do is speculate and relate to previous evidence, therefore I would change the word “confirms” to “suggests”.
R: Thank you for the suggestion. In this case, to support our results, it would make sense to cite longitudinal studies, to see if the results are confirmed over time. However, we have restructured the discussion, so we do not include this quote.
Line 245: In discussing the limitations of the study, please also make suggestions for future research regarding how these limitations could be addressed.
R:. We make some suggestions for future research.
Action:
Line 269 to 277: “First, besides underestimating upper body movements and activities such as carrying heavy loads, weight training, and cycling [55], accelerometers devices are not water resistant. Furthermore, they cannot capture postural information (i.e., sitting vs. standing still), which may lead to the overestimation of sedentary time [18]. Secondly, the cross-sectional study design does not allow for causal conclusions to be drawn. This in an important issue because behaviour patterns may reflect physical fitness. Future longitudinal or experimental studies are needed to adress this limitation. Also, it would be relevant to analyse not only physical activity quantity but also the type of physical activity performed (e.g., using the Physical Activity Questionnaire [56]).”
Line 246: With your second limitation, go a step further and explain what implications this limitation of the accelerometers is (i.e., they can underestimate total PA / miss-classify PA, etc).
R: Accelerometers can underestimate the physical activity performed with the upper body movements. They are also unable to capture postural information, which can lead to overestimation of sedentary time.
Action:
Line 269 to 272: “First, besides underestimating upper body movements and activities such as carrying heavy loads, weight training, and cycling [55], accelerometers devices are not water resistant. Furthermore, they cannot capture postural information (i.e., sitting vs. standing still), which may lead to the overestimation of sedentary time [18].”
Overall comment: What about power? Were you powered to detect significant associations? Also, I don’t believe the SFT specifically includes a measure of BMI. It’s an important variables but it should be made clear that this was an addition to the SFT.
R: A power analysis using the G*Power (3.1.9.2) computer program indicated that the total sample of 84 people would be needed to detect meddium effect (r= 0.3) (Cohen, 1988) with 80% power using the test correlation bivariate normal model, with alfa at 0.05. (Cohen, J. Statistical power analysis fot he behavioral sciences (2ª edition) New York: Lawrence Erlbaum Pub) .
Action:
Line 148: “A power analysis using the G*Power (3.1.9.2) computer program indicated that the total sample of 84 people would be needed to detect medium effect (r= 0.3) [34] with 80% power using the test correlation bivariate normal model, with alfa at 0.05.”
Regarding BMI is a variable that is part of the Senior Fitness Test battery, because of previous evidence showing its role in maintaining functional mobility (In: Rikli, J., and Jones. Development and validation of a functional fitness test for a community-residing adults. Journal of aging and physical activity 1999, 7(2),129-161)

Round 2
Reviewer 2 Report
I appreciate the authors’ efforts to revise the manuscript. The changes made have clearly improved the paper. However, I suggest that some minor changes need to be done before acceptance.
P 2 line 51 and 61: MVPA and LPA need to be spelled out the first time mentioned
P 2 line 57: I suggest to change “the accelerometer is a precise method” to “accelerometry is a more accurate method”
P 2 line 65: I am not sure what the authors mean by “physical fitness adjustments”?
P2 line 72: ...evidence shows that sitting for longer...
P 2 line 78: I suggest to change to “sedentary and physical activity in the elderly“
P 2 line 79: I suggest to change “verify” to “use accelerometry to examine” (also in abstract)
P 3 line 128: sedentary activity (<100 counts/min)
P 5 Table 1: There is still no information on number of accelerometer wear days
P 6 Table 3: Typo - Inactive group, Active group
P 6 line 205 and 206: No need to spell out MVPA and LPA.
P 8: It should also be mentioned among the limitations that this is a non-random sample and that the accelerometers were not worn for 7 days.
P 8 line 290: Future prospective studies using objective assessment of physical activity are required...
Author Response
Dear Dr. Reviewer,
My colleagues and I would like to thank you for the opportunity to resubmit our manuscript to the International Journal of Environmental Research and Public Health. We found that the reviewers’ comments were very helpful and we have done our best to incorporate all of their suggestions and reply to the reviewers` comments. We believe that this has made a significant contribution to the overall quality of the manuscript.
The reviewers’ comments and our actions are attached at the bottom of this letter. We have also included an updated version of our manuscript with all the changes highlighted in yellow.
If you require any additional information, please do not hesitate to get in touch with us.
Comments and Suggestions for Authors
I appreciate the authors’ efforts to revise the manuscript. The changes made have clearly improved the paper. However, I suggest that some minor changes need to be done before acceptance.
P 2 line 51 and 61: MVPA and LPA need to be spelled out the first time mentioned
Action:
Line 51: “week or an equivalent combination of MVPA performed in bouts of at least 10 minutes each” suggest changing to “week or an equivalent combination of moderate to vigorous physical activity (MVPA) performed in bouts of at least 10 minutes each”.
Line 62: “Despite that, LPA is also associated with physical fitness improvements in the elderly” suggest changing to “Despite that, light physical activity (LPA) is also associated with physical fitness improvements in the elderly.”
P 2 line 57: I suggest to change “the accelerometer is a precise method” to “accelerometry is a more accurate method”
R: Thank you for the suggestion.
Action:
Line 58: “the accelerometer is a precise method” suggest changing to “accelerometry is a more accurate method to assess both physical activity”.
P 2 line 65: I am not sure what the authors mean by “physical fitness adjustments”?
Action:
Line 66: “better physical fitness adjustments when compared to those who practiced MVPA” suggest changing to “better physical fitness results when compared to those who practiced MVPA”.
P2 line 72: ...evidence shows that sitting for longer...
R: Thank you for the suggestion.
Action:
Line 73: “being sat for longer” suggest changing to “evidence shows that sitting for longer than 4 hours/day”.
P 2 line 78: I suggest to change to “sedentary and physical activity in the elderly“
R: Thank you for the suggestion.
Action:
Line 79: “sedentary and MVPA times in the elderly, using objective measures” suggest changing to “sedentary and physical activity in the elderly, using objective measures”.
P 2 line 79: I suggest to change “verify” to “use accelerometry to examine” (also in abstract)
R: Thank you for the suggestion.
Action:
Line 18, 80 and 210: “The purpose of this study was to verify the relation between sedentary time, LPA and MVPA with the elderly’s physical fitness” suggest changing to “The purpose of this study was to use accelerometry to examine the relation between sedentary time, LPA and MVPA with the elderly’s physical fitness”.
P 3 line 128: sedentary activity (<100 counts/min)
Action:
Line 130: “(100 counts/min)” suggest changing to “(<100 counts/min)”.
P 5 Table 1: There is still no information on number of accelerometer wear days
Action:
Line 168: “a Accelerometry - 2 week valid days and 1 weekend valid day.” (Table 1)
P 6 Table 3: Typo - Inactive group, Active group
Action:
Table 3: “inative group” and “ative group” suggest changing to “inactive group” and active group”.
P 6 line 205 and 206: No need to spell out MVPA and LPA.
Action:
Line 210: “light physical activity (LPA) and moderate to vigorous physical activity (MVPA) with the elderly’s physical fitness” suggest changing to “LPA and MVPA with the elderly’s physical fitness.”
P 8: It should also be mentioned among the limitations that this is a non-random sample and that the accelerometers were not worn for 7 days.
R: Thank you for the suggestion. We include this limitation.
Action:
Line 278: “Also, this is a non-random sample and that the accelerometers were not worn for 7 days.”
P 8 line 290: Future prospective studies using objective assessment of physical activity are required...
R: Thank you. We modified the sentence and added more information.
Action:
Line 295: “Future prospective studies are, however, required to clarify causal relationships between physical activity levels and physical fitness among older people” suggest changing to “Future prospective studies using objective assessment of physical activity are required to clarify causal relationships between physical activity levels and physical fitness among older people. For the maximized benefits of physical activity, the elderly should be encouraged to interrupt the daily sedentary behaviour, avoiding long sitting periods”.

Reviewer 3 Report
Thank you for your extensive revisions to this manuscript. While the paper has been greatly improved, my biggest concern still is the novelty of this study. Your conclusions now match your findings, but I'd love to see some more critical thinking in how this body of research can be put into practice.
Specifically, I have a few additional comments:
Abstract:
Line 35: Here you say that MVPA was negative associated with agility/dynamic balance. While I understand that this is technically true, it implies that there is a detrimental association when in fact it is a beneficial one. I would consider revising this sentence.
Methods:
Line 104: The word "comitte" is spelt "committee".
Results:
Table 3: In your column headings you have incorrectly spelt the words "active" and "inactive". They are both missing the letter "C".
Discussion:
I would make it clear that your first 4 paragraphs discuss the results based on pooled data, not stratified by active/inactive.
Overall comment: While the main points that you are trying to convey in your manuscript can be understood, I would encourage you to get this paper professionally proof-read, given that you are submitting to an English-based journal.
Author Response
Dear Dr. Reviewer,
My colleagues and I would like to thank you for the opportunity to resubmit our manuscript to the International Journal of Environmental Research and Public Health. We found that the reviewers’ comments were very helpful and we have done our best to incorporate all of their suggestions and reply to the reviewers` comments. We believe that this has made a significant contribution to the overall quality of the manuscript.
The reviewers’ comments and our actions are attached at the bottom of this letter. We have also included an updated version of our manuscript with all the changes highlighted in yellow.
If you require any additional information, please do not hesitate to get in touch with us.
Reviewer 3 Comments
Comments and Suggestions for Authors
Thank you for your extensive revisions to this manuscript. While the paper has been greatly improved, my biggest concern still is the novelty of this study. Your conclusions now match your findings, but I'd love to see some more critical thinking in how this body of research can be put into practice.
R: For the public health prevention practices, it is of value to identifying modifiable behavioral factors associated with physical function among older adults in rapid aging societies. According to the literature, the relationship between sedentary time and physical function is less clear. Concerning physical activity, some studies show that higher levels of MVPA tended to be associated with better physical function, particularly in adults aged 65 years and older. However, it should be noted that, in the majority of these studies, physical activity duration and intensity were based on self-report methods which are known to significantly reduce data accuracy. Thus, we decided to examine the relationship between physical fitness and sedentary and physical activity in the elderly, using objective measures. This paper provides information for further studies to design physical activity intervention strategies for older adults with similar lifestyles. Sports and health professionals should reinforce the importance of preserving or improving physical fitness. That reinforcement can be achieved by recommending regular physical activity practice among the elderly (e.g., avoid transport and walking; climb stairs and avoid the elevator; engage in a regular exercise program).
Action:
Line 269: “Sports and health professionals should reinforce the importance of maintaining or improving physical fitness. That reinforcement can be achieved through recommending regular physical activity practice among the elderly (e.g., avoid transport and walk; climb stairs and avoid the lift; engage in a regular exercise program).”
Line 295: “Future prospective studies using objective assessment of physical activity are required to clarify causal relationships between physical activity levels and physical fitness among older people. For the maximized benefits of physical activity, the elderly should be encouraged to interrupt the daily sedentary behaviour, avoiding long sitting periods.”
Specifically, I have a few additional comments:
Abstract:
Line 35: Here you say that MVPA was negative associated with agility/dynamic balance. While I understand that this is technically true, it implies that there is a detrimental association when in fact it is a beneficial one. I would consider revising this sentence.
R: Thank you for the suggestion.
Action:
Line 28: “MVPA time was negatively correlated with body max index (BMI) [(rs=-0.218; p=0.048; -0.3< r ≤-0.1 (small)] and agility test performance [(rs=-0.367; p=0.001; -0.5< r ≤-0.3 (low)]” suggest changing to “MVPA time was correlated with lower body mass index (BMI) [(rs=-0.218; p=0.048; -0.3< r ≤-0.1 (small)] and shorter time to complete the agility test”.
Line 35: “MVPA time was negatively correlated with BMI [(rs=-0.320; p=0.020; -0.5< r ≤-0.3 (low)], and agility test/dynamic balance performance [(rs=-0.296; p=0.031; -0.3< r ≤-0.1 (small)] suggest changing to “MVPA time was correlated with lower BMI [(rs=-0.320; p=0.020; -0.5< r ≤-0.3 (low)], and shorter time to complete agility test.”
Methods:
Line 104: The word "comitte" is spelt "committee".
Action:
Line 106: “All procedures were approved by the local ethics Comitte” suggest changing to “All procedures were approved by the local ethics Committee”.
Results:
Table 3: In your column headings you have incorrectly spelt the words "active" and "inactive". They are both missing the letter "C".
Action:
Table 3: “inative group” and “ative group” suggest changing to “inactive group” and active group”.
Discussion:
I would make it clear that your first 4 paragraphs discuss the results based on pooled data, not stratified by active/inactive.
R: Thank you for the suggestion.
Action:
Line 232: “Concerning sedentary time, there also was no significant correlation with any variable under analysis.” suggest changing to “Still according to our pooled results, we also found that with sedentary time there was no significant correlation with any variable under analysis”.
Overall comment: While the main points that you are trying to convey in your manuscript can be understood, I would encourage you to get this paper professionally proof-read, given that you are submitting to an English-based journal.
R: Thank you for the suggestion. Our manuscript was checked by a native English speaking professional. We have attached the statement provided by the professional.
